# Colonzation of Tobacco Plants by Fungal Entomopathogens and the Effect on Consumption over *Diabrotica speciosa* (Coleoptera: Chrysomelidae)

**DOI:** 10.3390/jof7121017

**Published:** 2021-11-27

**Authors:** Florencia Vianna, Sebastian Pelizza, Leticia Russo, Natalia Ferreri, Ana Clara Scorsetti

**Affiliations:** Instituto Spegazzini, Universidad Nacional de La Plata, La Plata B1904, Argentina; florencia.vianna@fcnym.unlp.edu.ar (F.V.); sebastianpelizza@conicet.gov.ar (S.P.); nati_f@live.com.ar (N.F.); acscorsetti@fcnym.unlp.edu.ar (A.C.S.)

**Keywords:** fungal entomopathogens, endophytes, tobacco, biological control

## Abstract

Entomopathogenic fungi are naturally occurring microorganisms that attack insect pests, making them exceptional allies when developing biocontrol strategies. A particular aspect of the ecology of these fungi is that they interact not only with insects but also with plants, being able to colonize them endophytically without causing symptoms of disease. The objectives of this study were to determine the endophytic capacity of different species of entomopathogenic fungi in tobacco plants by means of foliar spraying, to evaluate the persistence in planta of the entomopathogens and the effect of endophytes on consumption by coleopteran pests. A total of 24 strains were analyzed to test endophytic capacity at 7, 14, 21 and 28 days post inoculation. A significant effect of the strains was found. On days 7, 14 and 21, the strain that showed the highest colonization percentages was *B. bassiana* LPSc 1215, and at day 28 *B. bassiana* strains LPSc 1215 and LPSc 1212 exhibited the best endophytic capacity, maintaining elevated colonization percentages. Choice test results indicated that *D. speciosa* fed indiscriminately on colonized and non-colonized leaves. The results indicate that *B. bassiana* LPSc 1215 constitutes a strain that would merit further investigation for the purpose of pest management in tobacco cultivation.

## 1. Introduction

Tobacco (*Nicotiana tabacum* L.) is an annual plant native to America that is currently grown worldwide. Despite the adverse effects associated with its consumption, it is a crop that constitutes the main source of income for the economies of certain regions, e.g., South America [1]. One of the major problems associated with its cultivation is that throughout the development cycle, chemical insecticide treatments are applied to control the appearance of insects that affect the leaf. Certain species of Lepidoptera and Coleoptera are particularly harmful since they cause serious yield losses and necessitate the use of large amounts of chemicals to mitigate their effects. These products negatively affect the environment and human health [2]. For years, integrated pest management programs (IPM) have sought to minimize chemical usage and employ biological control strategies instead, e.g., entomopathogenic fungi [3].

Entomopathogenic fungi are naturally occurring microorganisms that attack insect pests, making them exceptional allies when developing pest control programs. A particular aspect of the ecology of these fungi is that they interact not only with insects but also with plants, being able to colonize them endophytically without causing symptoms of disease [4,5]. As naturally occurring endophytes, entomopathogenic fungi have been found in few plant species; instead, investigations have focused on the possibility of artificially introducing them into plant microbiomes in order to provide protection against insect pests [6]. It has been suggested that, as endophytes, entomopathogenic fungi improve their biological control abilities since they do not have to face adverse environmental conditions and their compatibility with natural enemies increases [6,7]. Despite this advantage, the endophytic capacity of entomopathogenic fungi is highly variable, depending on the plant species, the fungal strain used and the inoculation method, among other factors [8]. *Beauveria bassiana*, *Metarhizium anisopliae* and *Purpureocillium lilacinum* (Ascomycota: Hypocreales) have been artificially inoculated and re-isolated with variable success from different crops, such as *Glycine max* [9], *Solanum lycopersicum* [10], *Gossypium hirstium* [11] and *Capsicum annum* [12]. In this respect, it is necessary to carry out particular studies on each system, observing the dynamics of the plant–fungus–insect interaction in order to evaluate the potential of entomopathogens to establish as endophytes and their potential to be incorporated in IPM programs as biocontrol agents.

Considering the previous background, the objectives of this study were to determine the endophytic capacity of different species of entomopathogenic fungi in tobacco plants, to evaluate the time persistence in planta of the entomopathogens and to evaluate the effect of endophytes on consumption by an insect plague.

## 2. Materials and Methods

### 2.1. Fungal Isolates

Fungal isolates were obtained from the Mycological Collection Carlos Spegazzini (LPSc), La Plata, Buenos Aires, Argentina (Table 1). Bioassays assessing growth rate, production of conidia and conidia viability were performed with 20 strains isolated from tobacco grown soils [13]. In order to test endophytic capacity, colonization experiments were performed with the previously mentioned strains and also with four strains obtained from the same mycological collection but with different origins (Table 1).

### 2.2. Growth Rate, Production and Germination of Conidia

The evaluation of the growth rate of the strains was performed at 10, 24 and 30 °C, following methods employed by Schapovaloff et al. [14]. Briefly, 5 mm diameter discs were obtained from 7 days old colonies grown on PDA (potato dextrose agar) (Britania^®^, Buenos Aires, Argentina) and were placed in the centre of 9 mm Petri dishes, containing PDA. The strains were incubated in darkness in a growth chamber for 14 days at 10, 24 and 30 °C. For every strain and temperature, five repetitions were made. The diameter of the colonies was measured daily using a caliper, considering two orthogonal axes to calculate the mean growth diameter. The radial growth was estimated in mm/day, using the following formula: growth rate (GR) = (final diameter/initial diameter)/incubation days.

After 14 days of incubation, conidia production and germination was estimated following Ayala Zermeño et al. [15]. To calculate conidia production, 10 mL of Tween 80 (Merck^®^, Kenilworth, NJ, USA) (0.01%, *v*/*v*) was added to Petri dishes and the surfaces of the colonies were gently scrapped with a sterile loop. The resulting suspensions were transferred to sterile test tubes that were vortexed for 5 min, and conidia production was recorded using a Neubauer chamber. In order to determine the germination percentage, 0.5 mL of PDA was placed in slides that were posteriorly inoculated with 50 μL of conidia solution with a concentration of 10^6^ conidia/mL and uniformly distributed with a Drigalski loop. The slides were kept in climatic chambers at 24 °C and darkness for 24 h. Conidial germination percentage was registered under light microscope. A total of 100 conidia on each of the three repetitions were recorded. The cells were considered germinated when the germinal tubes presented a length superior to the diameter of conidia [16].

### 2.3. Bioassay I: Endophytic Capacity

#### 2.3.1. Plants

*Nicotiana tabacum* L. seeds were provided by “Cooperativa de Tabacaleros de Jujuy Lta. Argentina”; the variety used was Virginia K394. The seeds were surface sterilized using a solution of 70% alcohol for 2 min and rinsed twice in sterile water. The substrate consisted of a mixture of earth–perlite–vermiculite in equal parts (1:1:1) and was tindalized for periods of an hour with intervals of 24 h over 3 consecutive days. The seeds were sowed in 276 cells and when the seedlings reached the two-leaf stage they were transplanted to 330 cm^3^ plastic pots. The plants were maintained in a greenhouse under controlled conditions (25 °C, 12 h L:D) and were watered as needed.

#### 2.3.2. Inocula and Leaf Aspersion Technique

The fungal colonies were grown in Petri dishes containing PDA under controlled conditions (24 °C and darkness) for 14 days to promote growth and sporulation. Ten mL of 0.01% Tween 80 (Merck^®^) was added to each dish and the solutions were obtained by gently scraping the surface of the dishes. The resulting conidia suspensions were filtered through sterile gauze to separate mycelium debris and were vortexed for 1 min. The conidia concentration was determined for each strain using a Neubauer Chamber and the suspensions were adjusted to 1 × 10^8^ conidia/mL according to Russo et al. [17].

The leaf aspersion technique was employed since it has been reported to be the most effective means of establishing fungal entomopathogens as endophytes in tobacco plants [9]. Plants were inoculated with each fungal solution at the stage of 4–6 true leaves. A manual atomizer was utilized to spray 2 mL of the inoculum over the abaxial surface of the leaves. Controls were treated in the same fashion but sprayed with a conidia-free solution of Tween 80 (0.01%). In order to prevent the inoculum reaching the substrate, the top of the pots was covered with alumina paper at the moment of application. Ten repetitions were made for each strain with their respective control treatments. The experiment was repeated three times. Plants were kept under uniform conditions of 25 ± 2 °C and 12 h L:D in a greenhouse.

#### 2.3.3. Fungal Re-Isolation

The endophytic capacity of the strains was recorded at 7, 14, 21 and 28 days post inoculation. The methodology followed was the one proposed by Russo et al. [17]. Ten treated plants and ten control plants for each strain were used for each period post inoculation. Plants were washed under running water and were separated into leaves, stems and roots. The plant organ surfaces were sterilized by dipping the plant material separately in a 70% alcohol solution for two minutes, then in sodium hypochlorite (commercial bleach 55 g Cl/l) for two minutes, after which they were rinsed three times in distilled sterilized water. Utilizing a sterile scalpel, leaves were cut into 1 cm^2^ squares and stems and roots into 1 cm segments. Six fragments of each organ were placed in 9 mm Petri dishes containing PDA supplemented with antibiotics. The accuracy of the sterilization procedure was assessed by placing 1 mL of the final rinse water in plates containing PDA. No record of growing microorganisms was observed. The plates were kept in climatic chambers under controlled conditions (25 ± 1 °C, in darkness). The emergence of the entomopathogens was weekly registered for a period of two months.

### 2.4. Bioassay II: Feeding Preference

#### 2.4.1. Insects

Adults of *Diabrotica speciosa* (Coleoptera: Chrysomelidae) were collected from horticultural crops located near La Plata city, Buenos Aires, Argentina (34°56′19.2″ S/58°06′3.8″ W). Adult insects were individualized in sterile plastic vials over a period of 20 days to discard any infections contracted in the field. During this period, fresh tobacco leaves and a sucrose solution were provided as food source.

#### 2.4.2. Choice Test

The effect of endophytic *B. bassiana* LPSc 1215 on the food preference of *D. speciosa* adults was established in a choice test. Experimental arenas were set up containing two tobacco leaf discs (25 mm diameter) placed over filter paper discs to prevent desiccation. One of the leaf discs corresponded to previous inoculated plants (7 days earlier) that were assessed for the presence of the entomopathogen as endophyte (as in fungal re-isolation) and the other to plants inoculated with an aqueous solution containing Tween 80 (control) [18,19].

Insects were released individually in the experimental arenas and were left for 24 h in a climatic chamber under controlled conditions (25 ± 2 °C, 14:10 L:D). After 24 h, leaf discs were removed and scanned to estimate the consumed foliar area using Image J software [20]. The entire bioassay was repeated three times with 30 replicates each. A *t*-test was applied to verify differences among treatments.

#### 2.4.3. Data Analysis

Data were tested for normality and homogeneity of the variance prior to statistical analyses. One-way ANOVA was performed to test for differences between strains in the growth rate, production and germination of conidia at different temperatures.

For each strain and each organ, percent colonization frequency was calculated using the Petrini and Fisher formula = (number of plant pieces showing fungal outgrowth/total number of plated plant pieces) × 100 [21]. Percentage values of plant colonization frequency were transformed to stabilize the variance. One-way ANOVA was performed to evaluate differences in percent colonization frequency between strains. Significant differences among strains (*p* < 0.05) were compared with Tukey’s test. To estimate colonization as a function of time, three models were tested (exponential, logistic and polynomial) for each organ and the whole plant for strains with the best endophytic capacity.

Student’s *t* test (*p* < 0.05) was used to compare the food preference of *D. speciosa*.

All data were analyzed using InfoStat version 2011 [22].

## 3. Results

*B. bassiana* colonies showed relatively rapid radial growth at 24 °C and 30 °C. Significant differences were recorded between strains at 10 °C (F: 19.73; df: 10; *p* < 0.0001), at 24 °C (F: 31; df: 10 *p* < 0.0001) and at 30 °C (F: 67.14; df: 10; *p* < 0.0001). (Table 2). The production of conidia was significatively different for *B. bassiana* strains at 10 °C (F = 24.53; df = 10; *p* < 0.0001), at 24 °C (F = 256.85; df = 10; *p* < 0.0001) and at 30 °C (F = 189.75; df = 10; *p* < 0.0001) (Table 2). The viability of the conidia was calculated as the percentage of germination (Table 2*)*. There were significant differences between the isolates in all cases (at 10 °C F = 4.8, df = 10, *p* = 0.0001; at 24 °C F = 9.52, df = 10, *p* < 0.0001; and at 30 °C F = 9.4, df = 10, *p* < 0.0001).

Mycelial growth of *P. lilacinum* colonies was relatively fast at 24 °C and 30 °C. The growth rate exhibited at different temperatures is shown in Table 2. Significant differences were recorded between the growth rates of the different strains at 10 °C (F: 4.26; df: 7; *p* = 0.002), at 24 °C (F: 50.21; df: 7 *p* < 0.0001) and at 30 °C (F: 137; df: 7; *p* < 0.0001). The production of conidia for the *P. lilacinum* strains was significatively different, at 10 °C (F = 30.36; df = 7; *p* <0.0001), at 24 °C (F = 347.23; df = 7; *p* < 0.0001) and at 30 °C (F = 251.14; df = 7; *p* < 0.0001) (Table 2). The viability of the conidia expressed as a percentage of germination did not show significant differences in any of the three temperatures evaluated (10 °C F = 2.1, df = 7, *p* = 0.07; 24 °C F = 1.52, df = 7, *p* = 0.19; and 30 °C F = 1.03, df = 7, *p* = 0.42). The germination percentages are shown in Table 2.

The *M. anisopliae* colony showed moderately rapid growth in PDA, reaching an average diameter of 8 ± 0 mm at 10 °C, 40.7 ± 1.5 mm at 24 °C and of 42.6 ± 3.5 mm at 30 °C. The respective growth rates at each temperature were 0.21 ± 0, 0.23 ± 0.01 and 0.27 ± 0.02 mm /day. The conidia production at 10, 24 and 30 °C was 2.8 × 10^5^, 5.18 × 10^7^ and 1.69 × 10^7^ respectively. The viability of the conidia was 98% ± 1.51, 99.2% ± 0.83 and 96.6 ± 2.07 at 10, 24 and 30 °C respectively.

### 3.1. Bioassay I

A total of 24 strains plus a control were analyzed to test endophytic capacity. To compare the strains, a one-way ANOVA was performed for each week (Day 7, 14, 21 and 28) considering the whole plant (root, stem and leaf). Colonization percentages are shown in Figure 1. For each date, a significant effect of the strains was found (Day 7: F = 74.19, d.f. = 24, *p* < 0.0001; day 14: F = 58.09, d.f. = 24, *p* < 0.0001; day 21: F = 48.3, d.f. = 24, *p* < 0.0001; day 28: F = 3.13, d.f. = 24, *p* < 0.0001), which was tested a posteriori using a Tukey test. On days 7, 14 and 21 the strain that showed the highest colonization percentages was *B. bassiana* LPSc 1215 (92%, 78% and 69% respectively). At day 28 after inoculation, *B. bassiana* strains LPSc 1215 and LPSc 1212 exhibited the best endophytic capacity, maintaining colonization percentages of 63% and 80%, respectively.

The polynomial model described colonization the best, indicating that the effect of time over colonization values is higher as time passes (Table 3).

### 3.2. Bioassay II: Endophytic Effect of B. bassiana on Food Preference of D. speciosa Adults

The average leaf areas consumed by *D. speciosa* are shown in Figure 2. There were no significant differences among the consumed areas of control and treated plants (t = 1.1; d.f = 50, *p* = 0.28), indicating that the insects fed indiscriminately on any leaf.

## 4. Discussion

*Beauveria bassiana*, *P. lilacinum* and *M. anisopliae* showed the fastest growth rates and conidia production at 24 °C. This has also been reported for other entomopathogenic fungi of these same species [23,24,25]. On the other hand, conidia germination was above 90% despite the temperatures measured. The three species of entomopathogens are cosmopolitan and are capable of growing in different geoclimatic regions. However, in this study, variation in growth parameters among isolates (all from the same province) was recorded, indicating that thermo-tolerance is an intrinsic feature of each particular strain regardless of its origin. Information about the thermal requirements of a strain will undoubtedly be useful and should be considered when selecting a potential microbial control agent for a particular region.

When seeking to establish an entomopathogen as an endophyte, there are several factors that can affect the experimental success of colonization. Among these are the inoculation technique, the concentration of the inoculum, biological aspects, such as the plant species, its stage of development and the conditions in which it is planted, as well as the fungal species inoculated [5]. The present study demonstrated that the foliar spray technique is effective in establishing entomopathogenic fungi as endophytes in tobacco plants. Foliar spraying is the most widely extended technique for experimental inoculation of fungal entomopathogens and has allowed these fungi to colonize several plant species [19,26,27]. In this study, the strain LPSc 1215, isolated from the soil in the Jujuy Province, Argentina, was the one that showed the highest endophytic capacity. The colonization values were high even at 28 days post inoculation.

The ideal situation when using fungal inoculation techniques is to achieve a systemic colonization of the microorganism; however, what usually happens is that the colonization is higher in that organ in which the conidia treatment is applied [27]. For example, Posada et al. [28] reported that in coffee plants foliar spraying favors colonization of leaves and that application to the soil results in greater colonization of the roots. In this study, the same pattern was observed for *B. bassiana* and *P. lilacinum*, since in general terms strains showed a greater capacity to colonize the leaves and stems of tobacco plants. On the other hand, *M. anisopliae* LPSc1366 was the only strain able to demonstrate endophytic capacity, colonizing roots preferentially. Different species of the genus *Metarhizium* are considered naturally linked to the rhizosphere [29], colonizing root cortical cells but not migrating to other plant organs [19,30]. Nevertheless, several authors have reported entomopathogen presence in leaves and stems in addition to the roots after foliar application [31,32]. A possible explanation for these is provided by Behie et al. [33], who explained that the differential colonization in roots, stems and leaves by endophytic fungi would reflect differences in the external environment and in the biotic conditions of the different organs. Based on these results, it could be established that the choice of inoculation method must be planned according to the objective (for example, the location of the pathogen to be controlled or the organ attacked by a certain herbivore).

The fact that it is possible to isolate different species of entomopathogenic fungi from different organs of tobacco plants, even though the inoculation is carried out on the leaf, indicates that these fungi can migrate throughout the plant. The differences in the colonization percentages between tissues could be due to microbiological and physiological differences between them and to the fact that each particular strain may present specificity for conditions obtaining in different organs [21].

The different species of entomopathogens showed different capacities to colonize tobacco plants. The strains of *B. bassiana* showed the highest frequencies of colonization, followed by those of *P. lilacinum*. In contrast, only the strain LPSc 1366 of *M. anisopliae* was able to endophytically colonize tobacco plants. This strain was isolated from tobacco farm soils, preferentially invading the roots of the plant. The ability of the different strains to achieve endophytic colonization seems to be an intrinsic characteristic of the genotype and should be evaluated prior to their selection as a control agent [6].

A decrease in the percentage of colonization over time was observed for most of the strains; the same was reported by Herrero et al. [34] for *Tolypocladium cylindrosporum*. Arnold et al. [35] stated that the mutualistic relationships that occur in the host between inoculated endophytes and those transmitted horizontally or vertically could affect the recovery of the strains of interest. This could be a possible explanation for what happens in tobacco plants, constituting a hypothesis that should be tested in the future.

Authors such as Lartey et al. [36] and Broza et al. [37] concluded that pest insects do not consume, or do so to a lesser extent, those plants that present different species of entomopathogenic fungi as endophytes. Additionally, Powell et al. [38] and Cherry et al. [39] observed that the damage caused by *Helicoverpa zea* (Lepidoptera: Noctuidae) and *Sesamia calamistis* (Lepidoptera: Noctuidae), respectively, was reduced when plants colonized with *B. bassiana* were offered to the insects. The same results were obtained by Mutune et al. [40] who observed that the feeding behavior of larvae of the dipteran *Ophiomyia* sp. was negatively affected by feeding them with treated plants of *Phaseolus vulgaris*. Lopez Castillo et al. [11] and Martinuz et al. [41] also demonstrated through preference tests that *Aphis gossypii* (Hemiptera: Aphididae) preferentially fed on non-colonized plants. However, in the present study this behavior was not observed, since no significant differences were recorded in the consumption of plant material by the adults of *D. speciosa* fed with the control plants and those that presented *B. bassiana* as an endophyte. Similar results were observed by Resquin-Romero et al. [31], who evaluated the weight of *Spodoptera littoralis* (Lepidoptera: Noctuidae) as a parameter, when fed with plants colonized with *Beauveria* sp. and *Metarhizium* sp. Likewise, Leckie et al. [42] and Lopez, Castillo and Sword [43] did not record an effect on the larval weight of *H. zea* fed with treated and control plants. It would be advantageous if negative effects on insect life cycle parameters after the consumption of colonized plant material were confirmed. Further studies should seek to provide answers to the question of whether endophytes exert indirect effects on the development and reproduction of plague species. Although these results provide novel information on the potential of entomopathogenic fungi as endophytes for pest control, the mechanisms involved in the interaction between *B. bassiana*, *D. speciosa* and tobacco plants remain to be elucidated.

## Figures and Tables

**Figure 1 jof-07-01017-f001:**
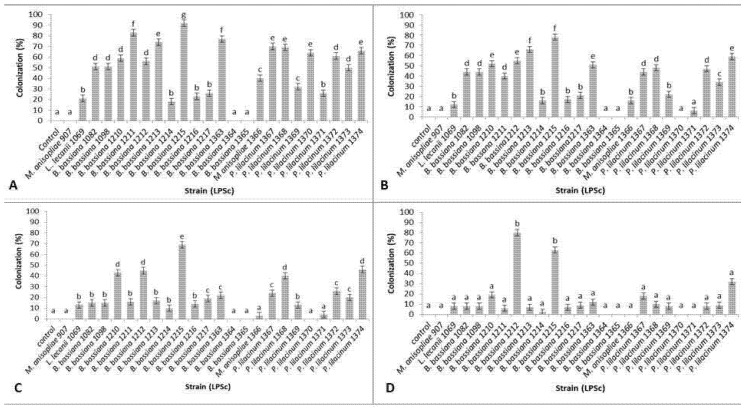
Endophytic colonization percentages (mean ± SE) of tobacco plants. (**A**) 7 days after treatment, (**B**) 14 days after treatment, (**C**) 21 days after treatment and (**D**) 28 days after treatment. Different letters indicate significative differences among strains on each date (Tukey test *p* < 0.05).

**Figure 2 jof-07-01017-f002:**
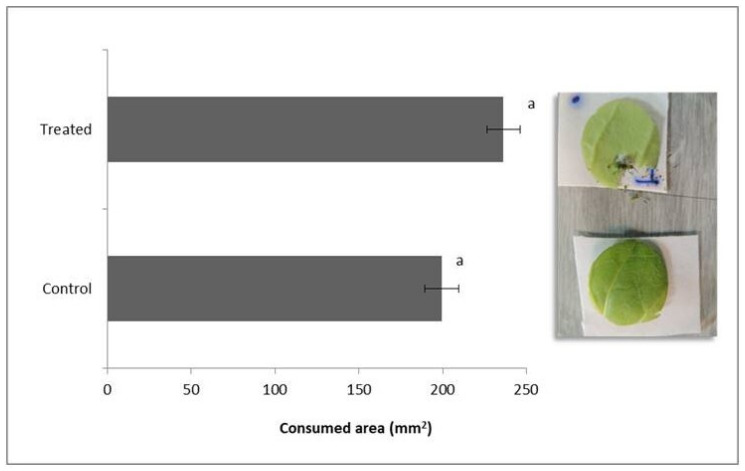
Leaf area consumed (mm^2^) by *Diabrotica speciosa* adults fed with *Beauveria bassiana* colonized (treated) and non-colonized (control) tobacco plants. Bars with the same letters are not significantly different according to a Student’s *t*-test (*p* < 0.05).

**Table 1 jof-07-01017-t001:** Strain origins of fungal entomopathogens.

Species	LPSc	ITS	TEF	Origin	Locality	Coordinates	Year of Isolation
*Beauveria bassiana*	1210	_	_	Soil	Perico, Jujuy	24°25′14.44″ S/65°01′14.81″ W	2014
	1211	MH050803	_	Soil	Perico, Jujuy	24°33′48.63″ S/65°04′58.79″ W	2014
	1212	_	_	Soil	Perico, Jujuy	24°25′14.44″ S/65°01′14.81″ W	2014
	1213	MH050801	MK047585	Soil	Perico, Jujuy	24°25′8.3″ S/65°01′26.8″ W	2014
	1214	MH050799	_	Soil	Perico, Jujuy	24°29′02.6″ S/64°58′21.7″ W	2014
	1215	MH050802	_	Soil	Perico, Jujuy	24°25′8.3″ S/65°01′26.8″ W	2014
	1216	MH050800	MK015641	Soil	Perico, Jujuy	24°29′02.6″ S/64°58′21.7″ W	2014
	1217	MH050798	_	Soil	Perico, Jujuy	24°29′50.8″ S/64°59′19.6″ W	2014
	1363	_	MK047587	Soil	Perico, Jujuy	24°33′37.1″ S/64°54′36.8″ W	2014
	1364	MH050805	MK047588	Soil	Perico, Jujuy	24°33′37.1″ S/64°54′36.8″ W	2014
	1365	_	_	Soil	Perico, Jujuy	24°29′02.6″ S/64°58′21.7″ W	2014
	1082	KJ7722495	_	Lepidoptera:Pyralidae	Tres estacas, Chaco	26°55′27″ S/61°37′36″ O	2009
	1098	KT163259	_	Hemiptera:Reduviidae	Tres estacas chaco, Chaco	26°55′27″ S/61°37′36″ O	2011
*Lecanicilium lecanii*	1069	_	_	Hemiptera: Aphididae	Concordia, Entre Rios	31°23′32″ S/58°01′01″ O	2009
	1367	MH050808	_	Soil	Perico, Jujuy	24°25′8.3″ S /W 65°01′26.8″	2014
	1368	MH050804	_	Soil	Perico, Jujuy	24°25′8.3″ S /W 65°01′26.8″	2014
	1369	MH050806	_	Soil	Perico, Jujuy	24°25′8.3″ S /W 65°01′26.8″	2014
	1370	MH050807	MK047586	Soil	Perico, Jujuy	24°25′8.3″ S /W 65°01′26.8″	2014
*Purpureocillium lilacinum*	1371	MK110011	MK047589	Soil	Perico, Jujuy	24°33′37.1″ S/64°54′36.8″ W	2014
	1372	_	_	Soil	Perico, Jujuy	24°29′50.8″ S/64°59′19.6″ W	2014
	1373	MH050809	MK047590	Soil	Perico, Jujuy	24°29′02.6″ S/64°58′21.7″ W	2014
	1374	MN516739	MK047591	Soil	Perico, Jujuy	24°29′02.6″ S/64°58′21.7″ W	2014
*Metarhizium anisopliae*	1366	_	_	Soil	Perico, Jujuy	24°29′02.6″ S/64°58′21.7″ W	2014
	907	KT163258	_	Hemiptera: Cercopidae	La Plata, Buenos Aires	34°47′26.45″ S/58°15′09.59″ W	2004

**Table 2 jof-07-01017-t002:** Strain growth rates, conidia production and germination at 10, 24 and 30 °C. Different letters among isolates of the same species (same row) indicate significant differences according to Tukey’s test (*p* < 0.05).

Strain		Growth Rate (mm/d)			Conidia Germination (%)		Conidia Production (Conidia/mL)		
**LPSc**	10 °C	24 °C	30 °C	10 °C	24 °C	30 °C	10 °C	24 °C	30 °C
*B. bassiana* 1210	0.71 ± 0.08 bc	1.8 ± 0.06 a	1.39 ± 0.15 ab	97.8 ± 1.3 bc	98.6 ± 1.14 b	98.6 ± 1.14	1 × 10^6^ ± 2.29 × 10^5^ fg	2.82 × 10^7^ ± 2 × 10^6^ c	1.6 × 10^7^ ± 2.4 × 10^5^ d
*B. bassiana* 1211	0.84 ± 0.05 cd	2.59 ± 0.14 cd	1.52 ± 0.09 b	97.8 ± 1.3 bc	99.2 ± 1.3 b	98.4 ± 1.52 c	1 × 10^5^ ± 3.5 × 10^4^ ab	3.67 × 10^7^ ± 1.1 × 10^6^ cd	1 × 10^7^ ± 6.2 × 10^5^ c
*B. bassiana* 1212	0.58 ± 0.06 a	2.49 ± 0.06 cd	2.01 ± 0.06 cd	96.6 ± 2.61 bc	97.4 ± 2.88 b	95.8 ± 2.39 bc	1 × 10^5^ ± 2.75 × 10^4^ abc	1.1 × 10^7^ ± 3.7 × 10^5^ b	3.4 × 10^6^ ± 5.5 × 10^5^ b
*B. bassiana* 1213	0.99 ± 0.05 e	1.66 ± 0.05 a	1.3 ± 0.1 a	96.6 ± 2.61 bc	97.6 ± 2.07 b	98.6 ± 1.14 c	1.87 × 10^6^ ± 3.3 × 10^5^ g	3.4 × 10^7^ ± 7.1 × 10^5^ cd	2.5 × 10^7^ ± 2.9 × 10^6^ def
*B. bassiana* 1214	0.71 ± 0.05 abc	1.94 ± 0.05 ab	2.13 ± 0.06 d	98 ± 1.58 bc	99.8 ± 0.45 b	98 ± 1.58 c	7.1 × 10^5^ ± 1.24 × 10^5^ efg	2.8 × 10^7^ ± 1.7 × 10^6^ c	2 × 10^7^ ± 1.5 × 10^6^ de
*B. bassiana* 1215	0.75 ± 0.04 bc	1.87 ± 0.47 a	1.48 ± 0.07 ab	99.2 ± 0.84 c	100 ± 0.00 b	99 ± 0.71 c	7 × 10^4^ ± 1.2 × 10^4^ a	8.1 × 10^7^ ± 5.7 × 10^6^ e	3.8 × 10^7^ ± 7 × 10^5^ g
*B. bassiana* 1216	0.9 ± 0.11 de	2.61 ± 0.04 d	2.05 ± 0.11 cd	95.2 ± 2.28 abc	97.4 ± 2.41 b	96.6 ± 2.88 bc	2.3 × 10^5^ ± 4.6 × 10^4^ bcd	2 × 10^6^ ± 1.7 × 10^5^ a	8.5 × 10^5^ ± 9.8 × 10^4^ a
*B. bassiana* 1217	0.63 ± 0.02 ab	2.51 ± 0.09 cd	2.01 ± 0.06 cd	97.4 ± 1.82 bc	96.2 ± 2.77 b	93.4 ± 3.21 ab	1.7 × 10^5^ ± 5.15 × 10^4^ abc	2.8 × 10^7^ ± 1.8 × 10^6^ c	4.2 × 10^6^ ± 2.9 × 10^5^ b
*B. bassiana* 1363	0.74 ± 0.05 bc	1.63 ± 0.05 a	1.42 ± 0.08 ab	89.6 ± 1.14 a	90.8 ± 0.84 a	90.6 ± 2.3 a	3.2 × 10^5^ ± 2.5 × 10^4^ cde	7.8 × 10^7^ ± 5.2 × 10^6^ e	3.3 × 10^7^ ± 1.4 × 10^6^ fg
*B. bassiana* 1364	0.83 ± 0.06 cd	1.69 ± 0.14 a	1.29 ± 0.1 a	99.4 ± 0.89 c	99.6 ± 0.55 b	99 ± 1 c	5.8 × 10^5^ ± 9 × 10^4^ def	3.9 × 10^7^ ± 9.6 × 10^5^ d	2.6 × 10^7^ ± 1 × 10^6^ efg
*B. bassiana* 1365	0.7 ± 0.06 ab	2.26 ± 0.04 bc	1.9 ± 0.09 c	92.6 ± 8.26 ab	96.2 ± 3.03 b	97.8 ± 1.92 c	8.4 × 10^5^ ± 1.3 × 10^5^ efg	2.7 × 10^7^ ± 2 × 10^6^ c	8.1 × 10^6^ ± 5.7 × 10^5^ c
*P. lilacinum* 1367	0.68 ± 0.04 ab	1.23 ± 0.22 a	2.1 ± 0.11 a	98.6 ± 1.14 a	97.2 ± 1.79 a	98 ± 1.22 a	1.3 × 10^6^ ± 1.3 × 10^5^ b	3.5 × 10^7^ ± 4 × 10^5^ b	4.1 × 10^7^ ± 3.4 × 10^6^ d
*P. lilacinum* 1368	0.89 ± 0.27 b	1.98 ± 0.76 bc	2.04 ± 0.38 a	97.2 ± 1.79 a	98.6 ± 1.67 a	98 ± 2.35 a	1.1 × 10^6^ ± 1.8 × 10^5^ ab	1.8 × 10^7^ ± 4.4 × 10^5^ a	3.8 × 10^7^ ± 9.3 × 10^5^ d
*P. lilacinum* 1369	0.76 ± 0.07 ab	2.36 ± 0.09 c	2.62 ± 0.11 c	96.2 ± 2.39 a	97.4 ± 2.07 a	97.4 ± 2.35 a	1 × 10^6^ ± 8.5 × 10^4^ ab	3.2 × 10^7^ ± 8.1 × 10^5^ b	3.2 × 10^6^ ± 3.6 × 10^5^ a
*P. lilacinum* 1370	0.55 ± 0.03 a	2.18 ± 0.19 bc	1.94 ± 0.09 b	98 ± 1.58 a	97.4 ± 2.7 a	98.6 ± 1.14 a	7.6 × 10^5^ ± 5.5 × 10^4^ a	5.4 × 10^7^ ± 1.1 × 10^6^ c	1.2 × 10^7^ ± 4.3 × 10^5^ b
*P. lilacinum* 1371	0.79 ± 0.06 b	4.4 ± 0.17 e	4.31 ± 0.14 d	98.8 ± 1.3 a	99.8 ± 0.45 a	98.8 ± 0.84 a	1 × 10^6^ ± 5.8 × 10^4^ ab	1.6 × 10^7^ ± 2.3 × 10^5^ a	2.2 × 10^7^ ± 6.2 × 10^5^ c
*P. lilacinum* 1372	0.76 ± 0.08 ab	1.73 ± 0.17 abc	1.85 ± 0.08 ab	96 ± 2.24 a	98.6 ± 1.14 a	96.2 ± 2.28 a	8.2 × 10^6^ ± 3.4 × 10^4^ a	5.7 × 10^7^ ± 2 × 10^6^ c	4.4 × 10^7^ ± 1.1 × 10^6^ d
*P. lilacinum* 1373	0.68 ± 0.04 ab	1.68 ± 0.08 ab	1.56 ± 0.13 a	95.8 ± 2.28 a	96.8 ± 1.92 a	97.2 ± 2.49 a	9.2 × 10^5^ ± 9.8 × 10^4^ ab	3.6 × 10^7^ ± 1.6 × 10^6^ b	4.9 × 10^7^ ± 9.1 × 10^5^ d
*P. lilacinum P. lilacinum* 1374	0.79 ± 0.08 b	3.34 ± 0.34 d	2.11 ± 0.04 b	97.8 ± 1.48 a	98.2 ± 1.64 a	98.6 ± 1.52 a	3.8 × 10^6^ ± 1.5 × 10^5^ c	5.9 × 10^7^ ± 9.7 × 10^5^ c	6.8 × 10^7^ ± 4 × 10^6^ e
*M. anisopliae* 1366	0.21 ± 0	0.23 ± 0.01	0.27 ± 0.02	98.4 ± 1.51	99.2 ± 0.83	96.6 ± 2.07	2.8 ×10 ^5^ ± 8 × 10^4^	5.18 × 10^7^ ± 1.47 × 10^6^	1.69 × 10^7^ ± 1.2 × 10^6^

**Table 3 jof-07-01017-t003:** Coefficients of the polynomial exponential second order models (a + b × time + c × time^2^) fitted to the colonization data of strains LPSc 1212 and LPSc 1215.

		a	b	c	R²	R
LPSc1212	Leaf	0.6115	0.024386	−0.001133	0.560	0.763
	Root	0.631875	−0.014739	−0.00011	0.778	0.888
	Stem	0.223625	0.040354	−0.00139	0.445	0.688
	Plant	0.489	0.016667	−0.000878	0.340	0.592
LPSc1215	Leaf	1.07625	−0.015679	0.000189	0.449	0.691
	Root	1.1255	−0.0374	0.000449	0.671	0.829
	Stem	1.04925	−0.027836	0.000505	0.332	0.605
	Plant	1.083667	−0.026971	0.000381	0.341	0.594

a: intercept. b, c: slope. R^2^: Coefficient of determination. R: Pearson correlation coefficient.

## Data Availability

Not applicable.

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
