# Peer review of "Colonzation of Tobacco Plants by Fungal Entomopathogens and the Effect on Consumption over Diabrotica speciosa (Coleoptera: Chrysomelidae)"

_jof, 2021, doi:10.3390/jof7121017_

Round 1

Reviewer 1 Report

Specific comments:

Title: Diabrotica speciosa should be in italics

Abstract: line 17 & 20- B. bassiana should be in italics

Introduction: line 25 – Nicotiana tabacum should be in italics; line 48 &50-51 – names of fungal and plants species should be in italics

Materials and methods:

There is no information that the species affiliation of all studied fungal isolates was confirmed by molecular tests. Line 87 - was the germination of spores surely examined with a stereoscopic microscope? Or rather a light microscope?

Results: Since as many as 24 strains of fungi have been studied, both in Table II and in Fig. 1, I propose to somehow mark the species affiliation of individual isolates, i.e. B. bassiana, L. lecanii, P. lilacinum and M. anisopliae. This information can be placed, for example, under the horizontal axis of the chart. This will increase the transparency of the presented data. In Table II, the values ​​for spore production and titer should be superscript. There is no explanation in the legend under Table III what the symbols a, b, c R 2 and R mean.

Author Response

Comments and Suggestions for Authors
-Specific comments:

-Title: Diabrotica speciosa should be in italics

Done

-Abstract: line 17 & 20- B. bassiana should be in italics

Done

-Introduction: line 25 – Nicotiana tabacum should be in italics; line 48 &50-51 – names of fungal and plants species should be in italics

Done

-Materials and methods:

There is no information that the species affiliation of all studied fungal isolates was confirmed by molecular tests.

Some of the strains were not studied by molecular methods since in a previous study (Vianna et al. 2020) we used ISSR markers and these strains were phylogeneticly related to other strains. Furthermore all of the strains employed in our study were determined by morphological features by experimented mycologists and deposited at a worldwide recognized collection at our University (FCNyM, UNLP)

Vianna, M. F., Pelizza, S., Russo, M. L., Toledo, A., Mourelos, C., & Scorsetti, A. C. (2020). ISSR markers to explore entomopathogenic fungi genetic diversity: Implications for biological control of tobacco pests. Journal of Biosciences, 45(1), 1-11.

-Line 87 - was the germination of spores surely examined with a stereoscopic microscope? Or rather a light microscope?

Sorry, that was a mistake, the germination of conidia was examined with a light microscope. The text was modified.

-Results: Since as many as 24 strains of fungi have been studied, both in Table II and in Fig. 1, I propose to somehow mark the species affiliation of individual isolates, i.e. B. bassiana, L. lecanii, P. lilacinum and M. anisopliae. This information can be placed, for example, under the horizontal axis of the chart. This will increase the transparency of the presented data.

Species affiliation of each strain has been addressed.

-In Table II, the values ​​for spore production and titer should be superscript. There is no explanation in the legend under Table III what the symbols a, b, c R 2 and R mean.

Changes to the respective Tables have been made.

Reviewer 2 Report

The manuscript described endophytic colonization of tobacco plants by entomopathogenic fungi, and its use in control of D. speciosa. Below there are a few comments I hope will help the authors improve the manuscript. Title: Colonzation > Colonization; italize Diabrotica speciosa l.11-14 - The objectives of this study were to determine the endophytic capacity of different species of entomopathogenic fungi in tobacco plants by foliar spray technique, to evaluate the time persistence in-planta of the entomopathogens. Finally, to evaluate the effect of endophytes on consumption by choice test on coleopteran pest. > The objectives of this study were to determine the endophytic capacity of different species of entomopathogenic fungi in tobacco plants by foliar spray, to evaluate the time persistence in-planta of the entomopathogens, and to evaluate the effect of endophytes on consumption of coleopteran pest. l.47-48, 50-51 - italize cientific names Table 1 - Some of the strains were not identified by ITS or TEF, then how can you determine the species identification? l.71 - A commercial PDA was used? Which one? l.72 - What is APG? l.74 - How long was the experiment? l.75 - caliber > caliper l.79 - Ayala Zermeño et al. [15] methods > Ayala Zermeño et al. [15] l.84 - I believe you meant PLACED, not COLLOCATED l.86 - Why were all the experiments performed in darkness? Both germination and conidiation would improve under a few hours of light l.168 - Radial growth of B. bassiana colonies, showed relatively rapid growth at 24 °C and 30 °C. > B. bassiana colonies showed relatively rapid radial growth at 24 °C and 30 °C. l.168-203 - I think the authors should include in the text only the best results, since the others are already in Table 2. Otherwise it gets too tiring. Also, how did L. lecanii behave? l.204 - There are two M. anisopliae strains in Table 1, but only one appear in Table 2. Why? Table 2 - Title is incomplete. Also, first results line are not results, it is still header. And I suggest to put some kind of identification, which isolates are B. bassiana, P. lilacinum l.214 - What parameters were used to decide these 24 isolates? Figure 1 - Mean ±SE endophytic colonization porcentages of tobacco plants > Mean ±SE Endophytic colonization porcentages (Mean ±SE) of tobacco plants Table 3 - Title is incomplete. Figure 2 - Put the photo besides the respective bar, it will be clearer l.241 - Where are the results of L. lecanii? Discussion - The relevance of inoculation method is extensively described, but such evaluation was not performed.

Author Response

Comments and Suggestions for Authors

The manuscript described endophytic colonization of tobacco plants by entomopathogenic fungi, and its use in control of D. speciosa.

Below there are a few comments I hope will help the authors improve the manuscript.

-Title: Colonzation > Colonization; italize Diabrotica speciosa

done

-l.11-14 - The objectives of this study were to determine the endophytic capacity of different species of entomopathogenic fungi in tobacco plants by foliar spray technique, to evaluate the time persistence in-planta of the entomopathogens. Finally, to evaluate the effect of endophytes on consumption by choice test on coleopteran pest. > The objectives of this study were to determine the endophytic capacity of different species of entomopathogenic fungi in tobacco plants by foliar spray, to evaluate the time persistence in-planta of the entomopathogens, and to evaluate the effect of endophytes on consumption of coleopteran pest.

done

-l.47-48, 50-51 - italize cientific names

done

-Table 1 - Some of the strains were not identified by ITS or TEF, then how can you determine the species identification?

Some of the strains were not studied by molecular methods since in a previous study (Vianna et al. 2020) we used ISSR markers and these strains were phylogeneticly related to other strains. Furthermore all of the strains employed in our study were determined by morphological features by experimented mycologists and deposited at a worldwide recognized collection at our University (FCNyM, UNLP)

Vianna, M. F., Pelizza, S., Russo, M. L., Toledo, A., Mourelos, C., & Scorsetti, A. C. (2020). ISSR markers to explore entomopathogenic fungi genetic diversity: Implications for biological control of tobacco pests. Journal of Biosciences, 45(1), 1-11.

-l.71 - A commercial PDA was used? Which one?

Yes, Britania. It has been added to the manuscript

-l.72 - What is APG?

Sorry. This has been a language mistake; APG is the Spanish abbreviation for PDA.

-l.74 - How long was the experiment?

The experiment lasted 14 days. It is specified in line 73

-l.75 - caliber > caliper

done

-l.79 - Ayala Zermeño et al. [15] methods > Ayala Zermeño et al. [15]

done

-l.84 - I believe you meant PLACED, not COLLOCATED

yes, thanks the word has been replaced

-l.86 - Why were all the experiments performed in darkness? Both germination and conidiation would improve under a few hours of light

This experimental condition is usually employed when working with fungi. The experiments were conducted according the cited literature in order to stimulate growth and conidiation.

-l.168 - Radial growth of B. bassiana colonies, showed relatively rapid growth at 24 °C and 30 °C. > B. bassiana colonies showed relatively rapid radial growth at 24 °C and 30 °C.

done

-l.168-203 - I think the authors should include in the text only the best results, since the others are already in Table 2. Otherwise it gets too tiring. Also, how did L. lecanii behave?

Thanks for the suggestion, we have now included only the best results. L. lecanii was not tested in this bioassay.

-l.204 - There are two M. anisopliae strains in Table 1, but only one appear in Table 2. Why?

  1. anisopliae strain 907 was not tested for its growth.

-Table 2 - Title is incomplete. Also, first results line are not results, it is still header. And I suggest to put some kind of identification, which isolates are B. bassiana, P. lilacinum

done

-l.214 - What parameters were used to decide these 24 isolates?

Twenty out of the24 strains were selected since are native to tobacco grown soils and have not been tested for their endophytic capacity before. The remaining 4 strains were tested in previous bioassays and were employed to contrast with the other ones.

-Figure 1 - Mean ±SE endophytic colonization porcentages of tobacco plants > Mean ±SE Endophytic colonization porcentages (Mean ±SE) of tobacco plants

Done

-Table 3 - Title is incomplete.

Title has been completed

-Figure 2 - Put the photo besides the respective bar, it will be clearer

done

-l.241 - Where are the results of L. lecanii?

This strain was not commented since colonization results were not transcendent.

-Discussion - The relevance of inoculation method is extensively described, but such evaluation was not performed.

The text has been modified according to the suggestion

Reviewer 3 Report

Dear Authors, please see the pdf file for my comments

Author Response

All revisions were incorporated according to the suggestions made by reviewer in the attached .pdf.

Round 2

Reviewer 2 Report

-

Reviewer 3 Report

Dear Authors is good now, accepted for publication. Please write correct  the title